# WOLF: ACCURATE VIDEO CAPTIONING WITH A WORLD SUMMARIZATION FRAMEWORK

## ABSTRACT

We propose *Wolf*, a WOrLd summarization Framework for accurate video captioning. Wolf is an automated captioning framework that adopts a mixture-of-experts approach, leveraging complementary strengths of Vision Language Models (VLMs). By utilizing both image and video models, our framework captures different levels of information and summarizes them efficiently. Our approach can be applied to enhance video understanding, auto-labeling, and captioning. To evaluate caption quality, we introduce CapScore, an LLM-based metric to assess the similarity and quality of generated captions compared to the ground truth captions. We further build four human-annotated datasets in three domains: autonomous driving, general scenes, and robotics, to facilitate comprehensive comparisons. We show that Wolf achieves superior captioning performance compared to state-of-the-art approaches from the research community (VILA1.5, CogAgent) and commercial solutions (Gemini-Pro-1.5, GPT-4V). For instance, in comparison with GPT-4V, Wolf improves CapScore both quality-wise by $55.6\%$ and similarity-wise by $77.4\%$ on challenging driving videos. Finally, we establish a benchmark for video captioning and introduce a leaderboard, aiming to accelerate advancements in video understanding, captioning, and data alignment.

## 1 INTRODUCTION

Video captioning is crucial as it facilitates content understanding and retrieval by providing accurate, searchable descriptions. It also provides pairwise data for effective training of foundation models for tasks like video generation, such as Sora (Brooks et al., 2024) and Runaway (Runway, 2024). However, generating descriptive, accurate, and detailed video captions remains a challenging research problem for several reasons: firstly, high-quality labeled data are scarce. Video captions from the internet can be faulty and misaligned and human annotation is prohibitively expensive for large datasets. Secondly, video captioning is inherently more challenging than image captioning due to the additional complexity of temporal correlation and camera motion. Existing captioning models (Hong et al., 2024; Zhang et al., 2023) struggle with temporal reasoning and fail to achieve accurate scene understanding. Thirdly, there is no established benchmark to measure captioning progress. Existing video QA benchmarks (Maaz et al., 2023) are often limited to short answers, making it difficult to measure hallucinations in detailed long captions. Fourthly, the correctness and completeness of the captions are crucial for safety-critical tasks. In the era of LLMs, text descriptions of scenarios used by embodied agents for planning and control become increasingly common (Mao et al., 2023a;b; Li et al., 2024; Ding et al., 2023). Consequently, a false or incomplete description of the scenario may lead to the decision-making module overlooking a critical object after training on such caption data, resulting in safety risks. For instance, missing the presence of a human in the vicinity of a vegetable-chopping manipulator can lead to an injury.

To handle these challenges, we introduce WOrLd summarization Framework (*Wolf*), a novel summarization captioning framework, along with a captioning metric CapScore, and the Wolf captioning benchmark with corresponding datasets. Unlike previous works that utilize a single model to generate captions, we propose to use multiple models to collaborate (Jiang et al., 2024), producing much more accurate captions. By leveraging multiple models, we can provide more fine-grained details while reducing hallucinations. We show that Wolf achieves superior captioning performance compared to state-of-the-art approaches from the research community (such as VILA (Lin et al., 2023c), Co-

gAgent (Hong et al., 2024)) and commercial solutions (such as Gemini-Pro-1.5 (Team et al., 2023), GPT-4V (OpenAI, 2023)). In summary, we have three main contributions:

1. We design the first world summarization framework **Wolf** for video captioning and introduce an LLM-based metric **CapScore** for evaluating the quality of captions. We have further verified that CapScore aligns with human evaluations. The results show that our method improves CapScore by a large margin.

2. We introduce Wolf benchmark and four human-annotated benchmark datasets. These datasets include autonomous driving, general scenes from Pexels, and robotics videos, along with human-annotated captions, referred to as the **Wolf Dataset**.

3. The code, data and leaderboard will be open-sourced and maintained [1]. Continuous efforts and improvements will be made to refine the Wolf Dataset, codebase, and CapScore. We hope that Wolf will raise awareness about the quality of video captioning, set a standard for the field, and boost community development.

## 2 RELATED WORKS

**Image Captioning.** Visual language models (VLMs) have shown rapid advancements, achieving leading performance in image captioning tasks, largely due to the success of large language models. CLIP (Radford et al., 2021) pioneered this field by training a shared feature space for vision and language modalities on image-caption pairs. Building on CLIP, BLIP (Li et al., 2022) and BLIP-2 (Li et al., 2023) improved performance by aligning the pre-trained encoder with large language models. Following the direction, LLaVA (Liu et al., 2023) and InstructBLIP (Dai et al., 2023) demonstrated that jointly training on diverse datasets as an instruction-following task leads to strong generalization across various tasks. VILA (Lin et al., 2023c) highlighted the importance of pre-training with diverse data, and therefore significantly scaled up the pre-training dataset. Kosmos-2 (Peng et al., 2023) and PaLI-X (Chen et al., 2023a) further introduced pseudo-labeling bounding boxes from open-vocabulary object detectors to scale up the size of pre-training dataset.

**Video Captioning.** As image-based VLMs are not specifically trained with video data, they are limited in describing details present in the video data. To improve video captioning, PLLaVA (Xu et al., 2024) builds on top of LLaVa and introduced a parameter-free pooling strategy to enhance the caption quality. Video-llava (Lin et al., 2023a) achieves state-of-the-art performance on several benchmarks by conducting joint training on images and videos, thereby learning a unified visual representation. Additionally, Video-LLama (Zhang et al., 2023) incorporates both video and audio into LLMs by introducing two Q-formers to extract features. Vid2seq (Yang et al., 2023) conducts large-scale pre-training with narrated videos for dense video captioning. Meanwhile, MV-GPT (Seo et al., 2022) employs an automated speech recognition (ASR) model to provide additional labeling for the videos.

**LLM-based Summarization.** Recently many works have found that it is efficient to summarize useful information using LLMs. For example, LLaDA (Li et al., 2024) can provide users with helpful instructions based on the user request and corresponding traffic rules in the desired location. OpenAI team finds re-captioning (Betker et al., 2023) via LLMs can be very helpful.

## 3 WOLF: CAPTIONING EVERYTHING WITH A WORLD SUMMARIZATION FRAMEWORK

We propose Wolf, which is an automated captioning summarization framework that adopts a mixture of experts approach to generate long, accurate, and detailed captions for videos. Figure 1 provides an overview of our framework. In this paper, we use CogAgent (Hong et al., 2024), GPT-4V (Mao et al., 2023a) to generating image-level captions, and use VILA-1.5 (Lin et al., 2023c), Gemini-Pro-1.5 (Team et al., 2023) to generate video captions.

**Chain-of-thought Summarization in Image-level Models.** As image-level models (image-based VLMs) have been pre-trained with a larger amount of data than video-level models (video-based VLMs), we first use image-based VLMs to generate captions. We design a Chain-of-thought program to obtain video captions from image-level models. As illustrated in Figure 1, we first split the video

---

[1]We also provide ethical statement and reproducibility in Appendix.

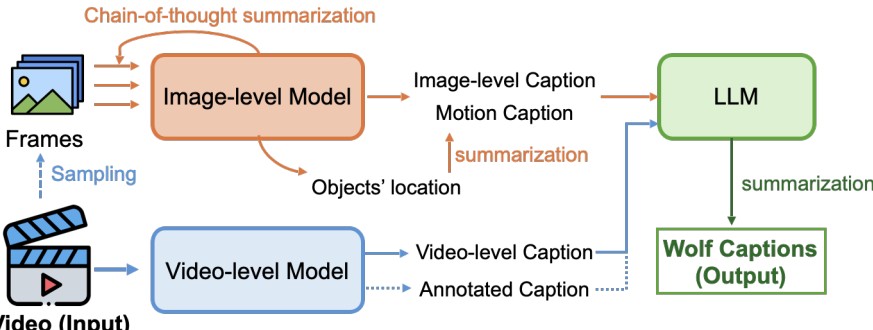

Figure 1: Overview of proposed Wolf framework. Wolf utilizes both image-level and video-level models to generate diverse and detailed captions, which are then summarized for cross-checking.

into sequential images, sampling two key-frames every second. We **start by** feeding Image 1 into the Image-level Model to obtain Caption 1, where we require the model to generate detailed scene-level information and object locations. Given the temporal correlation between key frames in a video, we then feed both Caption 1 and Image 2 into the model to generate Caption 2. By repeating this procedure, we generate captions for all sampled frames. Finally, we use GPT-4 to summarize the information from all captions with the prompt "*Summarize all the captions to describe the video with accurate temporal information*". Additionally, we extract the bounding box locations for each object in each frame, then feed them into LLMs to summarize the trajectory of the moving object. For example, in a driving video, a blue car is driving into the right lane, and the centers of the bounding boxes are (0,0), (1,1), (1,2). We provide the car's location to the LLM, and it outputs 'the blue car is driving to the right,' which we refer to as a **Motion Caption**.

**LLM-based Video Summarization.** Besides obtaining the captions from image-level models, we then summarize all captions into one. We use the prompt "*Please summarize on the visual and narrative elements of the video in detail from descriptions from Image Models (Image-level Caption and Motion Caption) and descriptions from Video Models (Video-level Caption)*". Optionally, we can also add the annotated caption to the summarization. Based on this simple scheme, Wolf can capture a rich variety of details of the video and reduce hallucinations (in Figure 2). We assume this is because the model can compare the captions and reduce redundant and hallucinated information. After obtaining the descriptions from the image-level and video-level models, we next apply the prompt "*Please describe the visual and narrative elements of the video in detail, particularly the motion behavior*".

## 4 WOLF BENCHMARK: BENCHMARKING VIDEO CAPTIONING

To showcase the effectiveness of Wolf, we constructed four distinct datasets. These include two autonomous driving video captioning datasets based on the open-sourced NuScenes (Caesar et al., 2019) dataset (Creative Commons Attribution-NonCommercial-ShareAlike 4.0 International Public License), a general daily video captioning dataset from Pexels [2], and a robot manipulation video captioning dataset from an open-source robot learning dataset (Padalkar et al., 2023). These benchmark datasets are tailored to assess the caption model's scene comprehension and its behavior understanding capabilities, both of which are vital for auto-labeling in embodied AI tasks. All captions were generated using a combination of ground truth information, rule-based heuristics, human labeling, and GPT-based rewriting.

### 4.1 BENCHMARK DATASET CURATION

#### 4.1.1 AUTONOMOUS DRIVING DATASET

High-quality captions of driving videos are crucial not only for training video generation models but also for training VLMs to interpret the dynamic traffic environment. The NuScenes dataset is a large-scale collection of driving videos designed to accelerate autonomous driving research. It features 1,000 annotated scenes from Boston and Singapore. Each scene consists of a 20-second driving video clip that provides an ego-centric view from the ego vehicle. We split each scene into 5-second segments and provide the corresponding captions. Our captions emphasize the high-level driving behavior of the ego vehicle to stress-test the scene understanding ability and the behavior

---

[2]https://www.pexels.com/

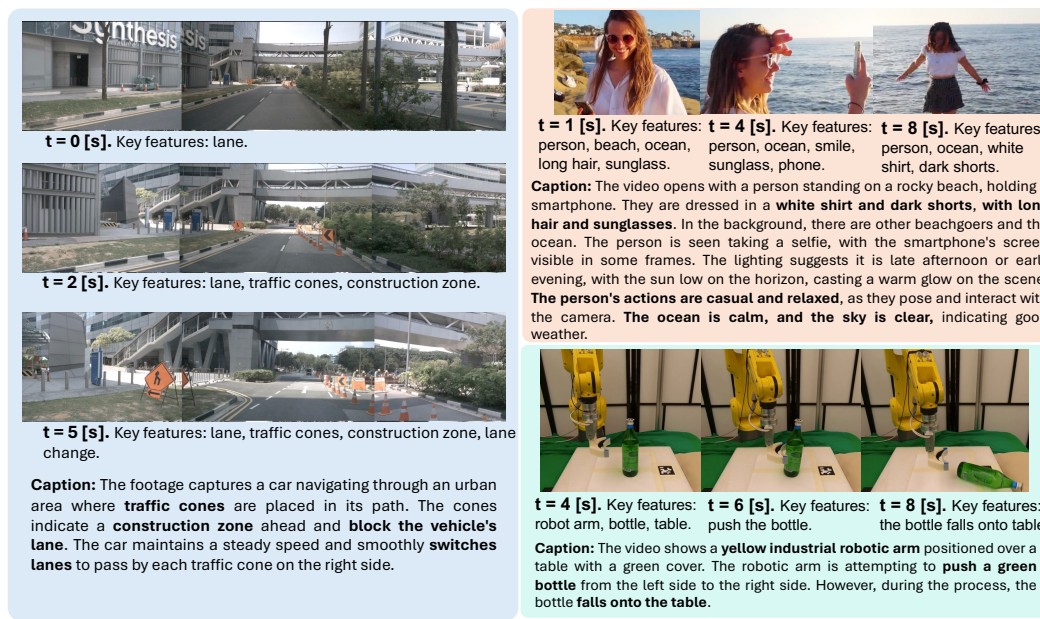

Figure 2: Wolf Dataset examples. We display the videos and corresponding human-annotated captions of autonomous driving (*Left*), Pexels (*Top-Right*), and Robot learning video dataset (*Bottom-Right*), totaling 25.7 hours for now, and the dataset size will be regularly updated and expanded.

understanding ability of the captioning model. Our dataset contains **500 intensely interactive video-caption pairs** (≈0.7 hours) in which the ego vehicle is involved in intense interactions with its surrounding traffic agents (such as navigating around construction zones and overtaking static obstacles) and **4785 normal driving scene video-caption pairs** (≈6 hours). Our caption generation process consists of three steps: i) agent-level motion annotation, ii) ego-centric interaction annotation, and iii) GPT-rewriting.

**Agent-level motion annotation.** The NuScenes dataset provides full annotation of the traffic elements in each scene, including the 3-D bounding box and categories of traffic elements, and semantic map information. Similar to Tian et al. (2024), we leverage this ground-truth information and the lane-topology information (Naumann et al., 2023) to annotate both the speed and angular motion characteristics of the ego vehicle and other traffic participants within a video clip. Specifically, we categorize agent actions into 11 types such as Stopping, Accelerating, Decelerating, Lane Changes, Turns, and more, based on their observed movements and behaviors.

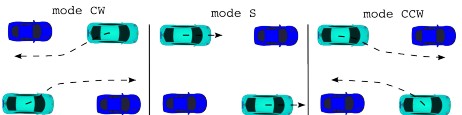

Figure 3: Illustration of homotopy types of different relative motions between a pair of vehicles.

**Ego-centric interaction annotation**. We are also interested in the ego vehicle's interaction with the other traffic participants (e.g., crossing pedestrians, blocking traffic cones, etc.) shown in the video clip. To efficiently caption the interaction, we leverage two types of categorical modes to describe the lane-relationship between a traffic participant and the ego vehicle (*agent-ego lane mode*) and the relative motion between a traffic participant and the ego vehicle (*homotopy*)(Chen et al., 2023b). Agent-ego lane mode at a time step $t$ encodes the topology relationship between the ego's current lane and the traffic agent's lane, including: *LEFT*, *RIGHT*, *AHEAD*, *BEHIND*, and *NOTON*, where *NOTON* describes that the traffic agent is not on any derivable lanes in the scene (e.g., a parked vehicle in a parking lot). To compute the agent-ego lane mode for each traffic agent, we follow (Chen et al., 2023b) to first identify the lane on which each agent is located and then leverage the lane topology map to annotate the agent-ego lane mode. We project the agent's center to the lane polyline and use its relative position in the local Frenet frame to determine its lane association. Homotopies describe the relative motion between a pair of agents shown in the video, including: [S, CW, CCW] (*static, clockwise, counterclockwise*), as shown in Figure 3.

**GPT-rewriting.** Combining agent-ego lane mode, homotopy, agent ground truth state information, and scene context information (e.g., ego is located near intersection) together, we can leverage

heuristics to annotate the interaction shown in the video clip. For example, in a video clip, a static object's agent-ego lane mode changes from *AHEAD*, to *LEFT*, to *BEHIND*, and the ego vehicle's first performs *RIGHT-LANE-CHANGE*, *KEEP-LANE*, then *LEFT-LANE-CHANGE*, indicating the ego vehicle overtakes that object from the ego vehicle's left side. We identified 6 interaction categories from the NuScenes dataset: 1) bypass blocking traffic cones to navigate around construction zone; 2) yield to crossing pedestrians; 3) yield to incoming vehicles; 4) overtake traffic agents via straddling the lane dividers; 5) overtake traffic agent via lane-change; 6) other non-intensive interactions. With both agent-level motion annotation and ego-centric interaction annotation, we use GPT 3.5 to summarize each clip to build the final caption.

### 4.1.2 ROBOT MANIPULATION DATASET

In addition to the driving environment, we collect **100 robot manipulation videos** (each has a length ranging from 5 seconds to 1 minute) from Padalkar et al. (2023) that demonstrate complex robot manipulations (e.g., pick and place, push, ect.) in various environments, including kitchen, office, lab, and open world. We manually caption each video. The captions focus on the description of the scene and the interaction between the robot and the objects (see the example in Figure 2).

### 4.1.3 PEXELS DATASET

To evaluate caption models in general daily environments, we further collect high quality (360p to 1080p) videos from Pexels [3]. It consists of **473 high-quality videos** sourced globally, where each video has a length varying between 10 seconds and 2 minutes and the content includes 15 popular categories (details in Appendix). This diversity not only adds depth to our dataset but also provides a wide range of scenarios and contexts for our analysis.

## 4.2 EVALUATION METRIC AND LEADERBOARD

### 4.2.1 CAPSCORE: EVALUATING CAPTIONS WITH LLMS

Video captioning has been an ill-posed problem since there is no metric to evaluate the quality of captions and the alignment between the video and the caption. Inspired by BERTScore (Zhang et al., 2019) and CLIPScore (Hessel et al., 2021), we introduce **CapScore** (Captioning Score), a quantitative metric to use LLMs to evaluate the similarity between predicted and human-annotated (ground-truth) captions. We tried both GPT-4 and LLama 3.1 (Dubey et al., 2024) as our LLM to summarize the captions. We noticed that GPT-4 can always obtain stable results over 3 runs. However, for LLama 3.1, the results varied over different runs. We tried to lower the temperature (from 0.9 to 0.5) to make the inference stable, however, we noticed that the scores are not consistent with human evaluation. Therefore we select GPT-4 as our LLM to conduct the experiments. Assume we have 6 captions, we feed all the captions into GPT-4 and add the prompt "*Can you give a score (two decimal places) from 0 to 1 for captions 1, 2, 3, 4 and 5, indicating which one is closer to the ground truth caption (metric 1) and which contains fewer hallucinations and less misalignment (metric 2)? Please output only the scores of each metric separated only by a semicolon. For each metric, please output only the scores of captions 1, 2, 3, 4 and 5 separated by commas, in order—no text in the output.*". We ask GPT-4 to output two scores: caption similarity and caption quality.

**Caption Similarity.** Caption similarity is based on how well each caption aligns with the ground truth description on a scale from 0 to 1, considering the key criteria mentioned. GPT-4 lists the requirements that affect the score: this metric measures how similar each caption is to the ground truth caption. The evaluation focuses on the content and context described in the captions, assessing whether they capture the main themes and details of the ground truth.

**Caption Quality.** Caption quality evaluates whether the caption contains reduced hallucination and mistakes compared to the ground truth captions on a scale from 0 to 1. GPT-4 lists the criteria that affect the score: this metric evaluates the accuracy and relevance of each caption, identifying any extraneous or incorrect details (hallucinations). Captions with fewer hallucinations and better alignment receive higher scores.

### 4.2.2 HUMAN-EVALUATION SCORE AND CAPSCORE

Through our experiments, we find that GPT-4 is very robust for calculating the scores. We have run the experiments for 1-3 times, the results appear to be stable and less than 0.05 changes. To alleviate

---

[3]https://www.pexels.com/

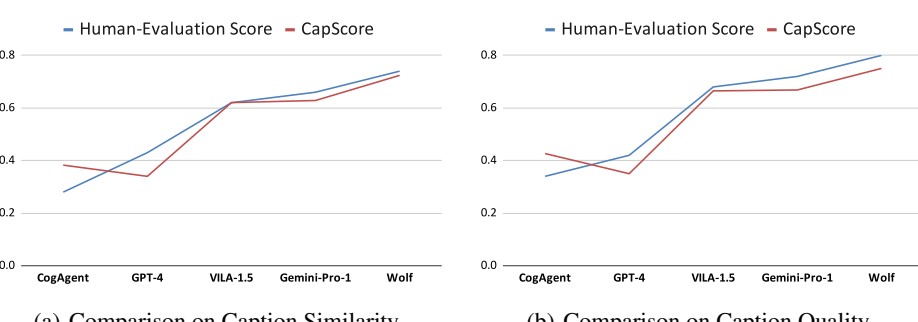

(a) Comparison on Caption Similarity.       (b) Comparison on Caption Quality.

Figure 4: Comparisons on Human-Evaluation Score and CapScore.

concerns related to human alignment and correlation, we randomly selected 10 users to evaluate our set of 100 robotics videos, as detailed in Table 1 of the paper. The evaluators were presented with the videos, the generated captions, and the corresponding ground truth captions. We asked them to assign human-evaluation scores based on the CapScore standard, with the following prompt: "*After reviewing the video and all the captions, please assign the caption similarity and caption quality score (floating point values) from 0 to 1 for different captions, indicating which caption is closest to the ground truth (caption similarity) and which one has fewer hallucinations and less misalignment (caption quality).*" We show the results in Table 1 the corresponding visual comparison in Figure 4.

| Method | Caption Similarity ↑ | | Caption Quality (eg. reduced hallucination) ↑ | |
| --- | --- | --- | --- | --- |
| | Human-Evaluation Score | CapScore | Human-Evaluation Score | CapScore |
| CogAgent (Hong et al., 2024) | 0.28 | 0.38 | 0.34 | 0.43 |
| GPT-4V (Achiam et al., 2023) | 0.43 | 0.34 | 0.42 | 0.35 |
| VILA-1.5 (Lin et al., 2023c) | 0.62 | 0.62 | 0.68 | 0.67 |
| Gemini-Pro-1.5 (Team et al., 2023) | 0.66 | 0.63 | 0.72 | 0.67 |
| **Wolf** | **0.74** | **0.72** | **0.80** | **0.75** |

Table 1: Comparison of Human-Evaluation Score and CapScore on 100 Wolf Robotics Videos.

### 4.2.3    BENCHMARKING VIDEO CAPTIONING

As far as we know, no standard evaluation benchmarks have been established for video understanding and captioning. To accelerate the advancement of this field, we have developed the first leaderboard for video captioning. As LLM evaluation has become increasingly popular (Chiang et al., 2024), we realized the lack of a standard platform to evaluate VLM's performance on video understanding. We assume this is due to the difficulty of collecting ground-truth captions that accurately align with videos. We will release the initial version of our captioning leaderboard upon publication.

## 5    EXPERIMENTS

### 5.1    EXPERIMENTAL SETUP

**Data Setup.** We use four sets of data to evaluate the validity of Wolf: 1) 500 Nuscences Interactive Videos; 2) 4,785 Nuscences Normal Videos; 3) 473 general videos and 4) 100 robotics videos. We extract 2 frames per second for autonomous driving videos. For robotics videos, we extract 1 frame per second. For short videos that sample less frames, we will increase `fps` to capture more details.

**Comparison Setup.** We use our proposed CapScore to evaluate the similarity between predicted and ground truth captions. CogAgent and GPT-4V are image-level methods, so we upload sequential frames into the model to obtain the output. VILA-1.5 and Gemini-Pro 1.5 are video-based methods, so we directly feed a video into the model. As for the prompt for each captioning model, we use "*elaborate on the visual and narrative elements of the video in detail, particularly the motion behavior*". We compare with four state-of-the-art image-level and video-level captioning method CogAgent (Hong et al., 2024), GPT-4V (Achiam et al., 2023), VILA-1.5 (Lin et al., 2023c) and Gemini-Pro-1.5 (Team et al., 2023). As for CogAgent, we feed the middle frame of the video into the model to obtain the captions. As for GPT-4V, we uniformly sample 16 frames from a video and feed the sequential images into the model to obtain captions. As for VILA-1.5 and Gemini-Pro-1.5, we feed the video into the model to obtain the captions.

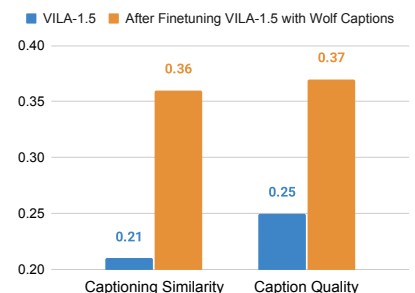

**Wolf** 🐺: The video presents a series of scenes depicting city streets bustling with various vehicles such as cars, trucks, motorcycles, and a person riding a bicycle. The scenes are often divided into panels, each showcasing a different vehicle in motion, including a yellow taxi, a blue car, a red car, a white truck, and a silver car. The streets are marked with white lines and are surrounded by green trees and buildings, contributing to an urban landscape setting. The video captures the typical flow of traffic through a four-way intersection, with vehicles **moving in different directions**. The vehicles are shown in various positions in the lanes, implying movement through the area. The sequence likely shows vehicles **accelerating** and **decelerating** as they approach and leave the intersection in response to traffic signals or the flow of other vehicles. The video is a wide-angle shot of the street intersection, split into three sections, providing a panoramic view. The left section shows the road the viewer is on, the center focuses on the intersection itself, and the right section provides a view of the crossing traffic. The lighting suggests a sunny day, with a bright spot from the sun on the left section and more even lighting on the right. The video captures the typical flow of traffic at an intersection, with vehicles approaching, some **waiting their turn while others proceed through the intersection when it's clear**. The cars driving on the right side of the road suggest the location might be a country where right-hand drive is the standard. The absence of sound enhances the focus on the visual information and allows the viewer to interpret the scene without auditory cues. Overall, the video provides a snapshot of everyday life at a **busy intersection**, highlighting the **complexities of navigating a traffic intersection**. The wide-angle perspective and focus on movement create a sense of dynamism.

Figure 5: Wolf example for driving videos that focus on interactive operations. Wolf captions discusses the motion behavior in details and serves as a good reference for autonomous driving.

| Method | Caption Similarity ↑ | | | Caption Quality (eg. reduced hallucination) ↑ | | |
|---|---|---|---|---|---|---|
| | Nuscenes | Pexels | Robotics | Nuscenes | Pexels | Robotics |
| CogAgent (Hong et al., 2024) | 0.18 | 0.68 | 0.38 | 0.24 | 0.72 | 0.43 |
| GPT-4V (Achiam et al., 2023) | 0.31 | 0.72 | 0.34 | 0.36 | 0.75 | 0.35 |
| VILA-1.5 (Lin et al., 2023c) | 0.21 | 0.85 | 0.62 | 0.25 | 0.86 | 0.67 |
| Gemini-Pro-1.5 (Team et al., 2023) | 0.42 | 0.87 | 0.63 | 0.45 | 0.87 | 0.67 |
| **Wolf** | **0.55** | **0.88** | **0.72** | **0.56** | **0.89** | **0.75** |

Table 2: Comparison on 500 highly interactive (difficulty and challenging) Nuscenes videos, 473 Pexels videos and 100 robotics videos. The best and second results are highlighted with **bold** and underline. Our Wolf exhibits better performance than both open- and closed-source models.

## 5.2 QUALITATIVE RESULTS

To illustrate enhanced captioning ability by Wolf, we show the qualitative results in Figure 5 (please check details in Appendix). We noticed that although GPT-4V is good at recognizing the scenes, capturing temporal information in a video is not ideal. Gemini-Pro-1.5 can capture video information such as "waiting their turn while others proceed through the intersection when it's clear", but it fails to describe the detailed motions. In comparison to these two state-of-the-art approaches, we observed that Wolf not only captures the motion described in Gemini-Pro-1.5 but also successfully captures "vehicles moving in different directions" and "vehicles accelerating and decelerating as they approach and leave the intersection in response to traffic signals or the flow of other vehicles".

## 5.3 QUANTITATIVE RESULTS

We compare Wolf with various state-of-the-art captioning models and display the results on 4 datasets in Table 2 and 3. In the default setting, Wolf uses CogAgent, GPT-4V, VILA-1.5, and Gemini-Pro-1.5 as Video-level models. Due to the running cost, we use Wolf (based on VILA-1.5) on the Nuscenes Normal dataset, which only uses CogAgent and VILA-1.5. We notice that existing image-level models fail to capture the temporal information in detail. Video-level models perform better, while Wolf can achieve the best results compared to all state-of-the-art captioning models.

Figure 6: Comparison between VILA-1.5 and fine-tuned VILA-1.5 with Wolf provided captions. on 500 highly interactive Nuscenes videos.

| Method | Caption Similarity ↑ | Caption Quality (eg. reduced hallucination) ↑ |
|---|---|---|
| CogAgent (Hong et al., 2024) | 0.27 | 0.30 |
| VILA-1.5 (Lin et al., 2023c) | 0.35 | 0.39 |
| **Wolf** (based on VILA-1.5) | **0.56** | **0.60** |

Table 3: Comparison on 4,785 normal Nuscenes videos. The quality of Wolf is consistently better.

| Method | Caption Similarity ↑ | Caption Quality (eg. reduced hallucination) ↑ |
|---|---|---|
| CogAgent | 0.18 | 0.24 |
| Wolf CogAgent part (chain-of-thought) | **0.26** | **0.32** |
| Wolf (based on VILA-1.5) | 0.35 | 0.37 |
| Wolf (based on VILA-1.5+Gemini-Pro-1.5) | 0.48 | 0.49 |
| Wolf (based on VILA-1.5+Gemini-Pro-1.5+GPT-4V) | **0.55** | **0.56** |

Table 4: Ablation study on 500 highly interactive Nuscenes videos.

### 5.4 FINETUNING VIDEO CAPTIONING MODELS

To further verify the effectiveness of Wolf, we finetune VILA-1.5 based on Wolf's captions on 4,785 normal Nuscenes videos and evaluate it on 500 highly interactive Nuscenes videos, which have much more difficult captions and complex scenarios. We follow the original VILA's training setup and launch supervised-finetuning with Wolf generated video-caption pairs for one epoch. The training is performed on 8xA100 GPUs with batch size 8. We set the learning rate to $10^{-4}$ with warmup strategy. No weight decay is applied.

We demonstrate the results in Figure 6, corresponding to Table 2. We observe that finetuning with Wolf boosts the model performance to 71.4% on caption similarity and 48.0% on caption quality, which outperforms GPT-4V and approaches Gemini-Pro-1.5. This suggests that Wolf captions can be easily applied to push VLMs' performance to a higher level.

### 5.5 ABLATION STUDY ON VIDEO-LEVEL MODEL SELECTION

To further evaluate how various video-level models affect the performance, we conduct an ablation study on the components of the models in Table 4. We first compare the caption from the middle frame of CogAgent with Wolf CogAgent Caption based on the chain-of-thought approach. The chain-of-thought procedure could largely improve the video understanding quality from an image-level model such as CogAgent. Then we compare Wolf with various combinations of video captions. We notice that Wolf consistensly shows better CapScore as it incorporates additional video captions.

### 5.6 COMPARISON OF FINETUNED MODELS

While it is difficult to directly and scalable measure the quality of captions, we compare the same model (VILA-1.5-13B) trained w/ Wolf captions and w/o Wolf captions to study the effectiveness. We benchmark the WOLF-finetuned models on two widely used video datasets ActivityNet (Caba Heilbron et al., 2015) and MSRVTT (Xu et al., 2016) and display the results in Table 5.

### 5.7 ABLATION STUDY ON TOKEN EFFICIENCY

It is well-known that the LLMs finetuned with RLHF favor longer response (Singhal et al., 2023), a phenomenon referred to as verbosity issue. To better assess the efficiency of the captions, we performed additional evaluation using the CapScore judge. Specifically, we separate each caption result into sentences, then incrementally use more sentences to form shortened captions, starting from only using the first sentence, to the whole original caption. These shortened captions are scored via CapScore, and we plot the score against the number of tokens used. We show the results in Figure 7.

From the result, we observe that for the better performing models (Wolf, Gemini-Pro-1.5 and GPT-4V) the similarity scores grow with token length when caption lengths are short, but quickly plateau or even drop as the caption lengths get too long. The caption quality score demonstrates quite diverse patterns from different models. GPT-4V maintains a relatively consistent quality score while Gemini-Pro-1.5 and Wolf display better quality when the caption length is short.

| | ActivityNet | MSRVTT |
|---|---|---|
| VILA-1.5-13B | 54.7 | 60.2 |
| VILA-1.5-13B (fine tuned with Wolf) | **55.2** | **60.9** |

Table 5: QA Accuracy Comparison of fine-tuned model on Activity and MSRVTT datasets.

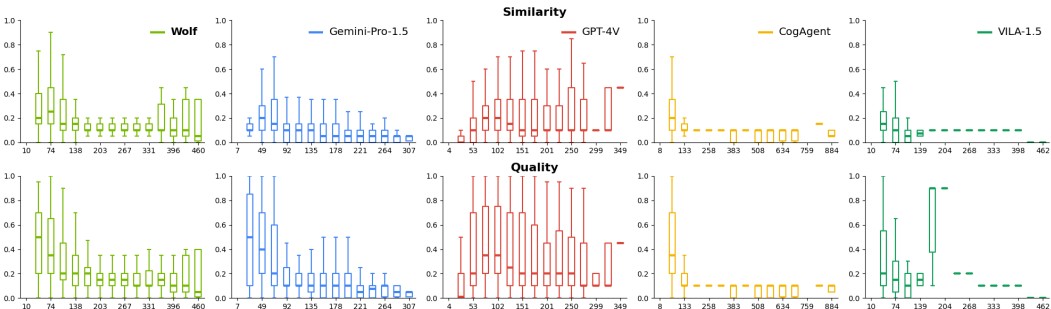

Figure 7: Caption Similarity / Quality evaluated by GPT-4 under varying caption length.

## 6 DISCUSSION AND FUTURE WORKS

**Limitations and Optimization.** Wolf is still significantly more cost-effective for autolabeling and captioning than procuring human labels. However, there is an efficiency concern when using an ensemble method like ours. This must be handled with great care to ensure that GPU resources are used effectively to mitigate any throughput degradation compared to using single models, even though Wolf offers a significant improvement in caption quality. Modern GPUs are based on a massively parallel pipeline, and our goal is to saturate this pipeline with meaningful work. We consider three primary areas for optimization to make Wolf a unified and efficient framework: Low-Hanging Fruit, Batched Inference, and Model Quantization. For example, we reduce the size of the model weights for model quantization. Several recent works (Lin et al., 2023b; Dettmers et al., 2024; Ma et al., 2024) have noted that LLMs and VLMs can produce highly accurate results even when their weights are quantized to low bit depths. Therefore, we quantize all constituent models used in Wolf to 4 bits to further improve efficiency. This has two benefits. First, it reduces the bandwidth required for computation. These algorithms work by packing two 4-bit numbers into a single 8-bit type, so when moving data on the GPU, only half the number of bits need to be moved. Since all currently released GPUs support native instructions on 8-bit floating point numbers, the two 4-bit numbers are extracted and expanded by each kernel. In other words, two computations can be performed for every move operation. Next-generation GPUs will natively support 4-bit data types, and we expect further efficiency improvements from having dedicated 4-bit multiply and add instructions. Second, it synergizes with batched inference since the model weights, which are traditionally 16-bit, now only require one quarter of the GPU memory they would ordinarily use. This allows us to fit larger batch sizes on each GPU and process more videos in parallel. Please check our appendix for details.

**Safety Considerations.** As an ensemble of captioners, Wolf mitigates the possibility of missing out on crucial information in the captions and rectifying any hallucinations that do not agree with the output of most models, which is a fundamental pillar for developing safe autonomous systems, as specified in the functional safety standard ISO 26262 (ROHM). Beyond the benefits of Wolf, there are still various open questions pertaining to safety of VLM captioners in deployment which we aim to explore more in future: (i) We need to align the captions with the task at hand; e.g., in a driving scenario, a detailed description of the foliage around the road, even if correct, is irrelevant and can potentially act as distractor for the decision maker. (ii) Complementary to the first point, we need to *measure* how well a caption aligns with the task at hand and develop an advanced version of CapScore. (iii) Finally, we need an approach to quantify the confidence we have in the captions by leveraging techniques from learning theory, such as conformal prediction (Shafer & Vovk, 2008). Most prior work in this direction assumes an MCQ-styled outputs or those where a unique correct answer exists (Ren et al., 2023; 2024), but these approaches do not translate to free-form text descriptions.

## 7 CONCLUSION

In this work, we propose Wolf, a captioning framework designed to automatically and accurately annotate any video, with significant improvements in data alignment. We find out that adopting a mixture of captioning models and summarization can largely boost the quality of the captions. This enables obtaining long, detailed, and accurate video captioning. Beyond that, we set up a leaderboard to boost the development of video captioning, which preserves a guarantee for data alignment. We will also set up a thorough library that contains different types of videos with high-quality captions, regional information such as 2D or 3D bounding boxes and depth, and multiple object motions.

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
