# WOLF: ACCURATE VIDEO CAPTIONING WITH A WORLD SUMMARIZATION FRAMEWORK (APPENDIX)

## A CONTRIBUTIONS

We would like to list Wolf Contributions:

**1) Framework and Evaluation Metric.** We designed a novel world summarization framework, **Wolf**, for video captioning and introduced an LLM-based metric, **CapScore**, to evaluate the quality of captions. The results show that our method significantly improves CapScore.

**2) Datasets and Benchmark.** We introduce the Wolf benchmark (leaderboard) and four human-annotated benchmark datasets. These datasets include autonomous driving, general scenes from Pexels, robotics videos, and human-annotated captions, collectively referred to as the **Wolf Dataset**.

**3) Intended Uses.** We believe Wolf can serve as one of the best practices (auto-labeling tool) for creating and curating paired datasets and benchmarks.

**4) Hosting, licensing, and maintenance plan.** The code, data, and leaderboard will be open-sourced and maintained. Continuous efforts will be made to refine the Wolf Dataset, Wolf codebase, and CapScore. We hope that Wolf will raise awareness about the quality of video captioning, set a standard for the field, and boost community development.

## B PEXEL DATASET CATEGORIES

We categorize videos from pexel into the following types: Travel & Events, Sports, Education, Pets & Animals, People & Blogs, Nonprofits & Activism, News & Politics, Music, Science & Technology, Comedy, Entertainment, Film & Animation, Gaming, Robotics, How to Styles.

## C REPRODUCIBILITY CHECKLIST

**Datasets Curation.** We have provided dataset curation details in "Sec 4.1 Benchmark Dataset Curation" of the main paper. The main contribution of the Wolf dataset is its human-annotated captions. Wolf is built upon existing data from Nuscenes, videos from the Pexels website (publicly available, downloaded, and used for free), and robot manipulation videos collected from the Open X-Embodiment dataset (Padalkar et al., 2023) (under a CC License). Since all the videos have already been or will be released publicly, we can directly follow the instructions to download and use them. We will continually increase the size and update the data information in these folders.

**Evaluation Procedures.** We have provided evaluation procedures in "Sec 4.2.1 CapScore: Evaluating Captions with LLMs" of the main paper. Since CapScore compares captions from different methods simultaneously, it can guarantee the quality of the evaluation. In our paper, we ran all the experiments three times and took the average score in the reported tables and figures of the paper.

## D ETHICS STATEMENT

We commit to the ICLR Code of Ethics and affirm that our work follows the rules for experimentation. We welcome any related discussions and feedback.

## E    Qualitative Comparison on Interactive Nuscenes Driving Videos

We display the details of Figure 5 of the paper (Wolf example for driving videos that focus on interactive operations) in Figure 1.

## F    Wolf Efficiency Optimization

We consider three primary areas: **Low-Hanging Fruit**, **Batched Inference**, and **Model Quantization** as optimizations which make Wolf a unified and efficient framework. Using the optimizations detailed in this section we were able to increase the speed of CogVLM by a factor of approximately 10x (450s/video to 41s/video), VILA throughput was similarly improved to only about 3s per video.

**Low-Hanging Fruit.** These are primarily systems concerns and work arounds for simplistically written APIs. For example, the off-the-shelf CogVLM (Hong et al., 2024) and VILA (Lin et al., 2023b) supporting code is heavily based on loading PIL images to present to a huggingface pipeline (Wolf et al., 2019). In the naive pipeline, videos would need to be decoded and then converted to PIL images before input to the respective pipelines, which in turn convert them to GPU PyTorch (Ansel et al., 2024) tensors. This is extremely inefficient. Instead, we can leverage the hardware video decoder present in modern GPUs to decode the videos directly to GPU tensors and rewrite the preprocessing pipelines to operate on these tensors directly. This has the additional benefit of shifting preprocessing transform work from CPU to GPU.

**Batched Inference.** Simplifying Wolf into the simplest terms, we are essentially performing repeated neural network inference. Surprisingly, most VLM supporting code is designed to run inference on only a single example at a time. However, just as in other deep-learning problems, there fundamentally no reason why we cannot processes multiple videos at a single time in batches. This step is crucial to maximizing the use of GPU resources. Processing a single example may only use as little as 25% of a modern datacenter GPU which would either increase the time to process a dataset or the number of GPUs required to complete a task in a fixed time budget. We can reimplement more of the supporting code to enable processing batches of as many videos as will fit in GPU memory at a single time yielding a linear speedup in processing. For example, if we can fit batches of 4 in GPU memory we observe a speedup of 4x over processing single examples.

**Model Quantization.** The final optimization we consider is to reduce the size of the model weights. Several recent works (Lin et al., 2023a; Dettmers et al., 2024; Ma et al., 2024) have noted that LLMs and VLMs can produce highly accurate results even when their weights are quantized to low bit-depths. Therefore, we quantize all constituent models used in Wolf to 4-bits to further improve efficiency. This has two benefits. First, it reduces the bandwidth required for computation. These algorithms work by packing two 4-bit numbers into a single 8-bit type, so when moving data on the GPU only half the number of bits need to be moved. Since all currently released GPUs support native instructions on 8-bit floating point numbers, the two 4-bit numbers are extracted and expanded by each kernel. In other words, two computations can be performed for every move operation. Next generation GPUs will natively support 4-bit datatypes and we expect further efficiency improvements from having dedicated 4-bit multiply and add instructions. Next, it synergizes with batched inference since the model weights, which are traditionally 16-bit, now only require one quarter of the GPU memory they would ordinarily use. This allows us to fit larger batch sizes on each GPU and process more videos in parallel.

## G    Updated Results and Documentation

We will release our webpage and the initial version of our captioning leaderboard upon publication. We will regularly update Wolf results and documentation on our webpage.

## References

Jason Ansel, Edward Yang, Horace He, Natalia Gimelshein, Animesh Jain, Michael Voznesensky, Bin Bao, Peter Bell, David Berard, Evgeni Burovski, Geeta Chauhan, Anjali Chourdia, Will Constable, Alban Desmaison, Zachary DeVito, Elias Ellison, Will Feng, Jiong Gong, Michael Gschwind, Brian Hirsh, Sherlock Huang, Kshiteej Kalambarkar, Laurent Kirsch, Michael Lazos, Mario Lezcano, Yanbo Liang, Jason Liang, Yinghai

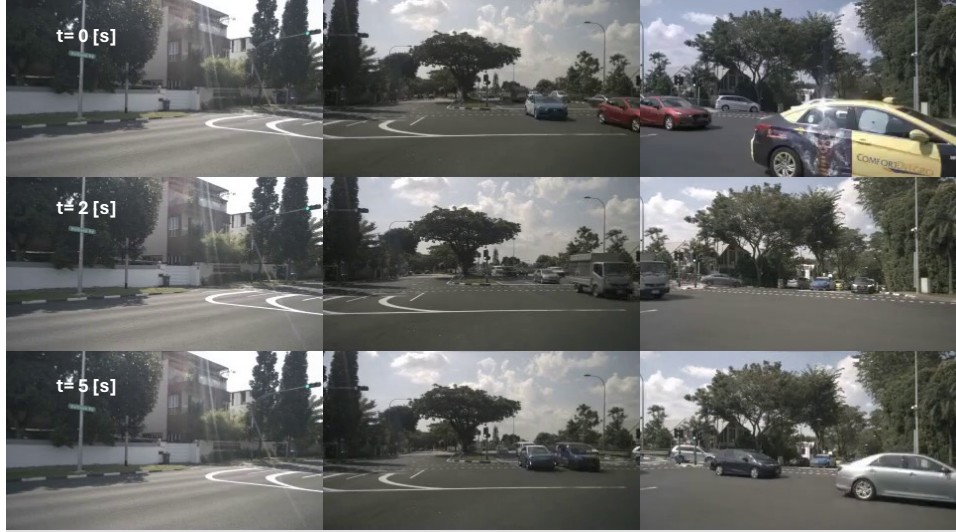

**CogAgent**: The picture shows a three-panel scene of a city street. In the first panel, a white truck is driving on the road, and there are several cars and motorcycles on the street. In the second panel, the white truck is still driving on the road, and there are several cars and motorcycles on the street. In the third panel, the white truck is driving on the road, and there are several cars and motorcycles on the street.

**VILA-1.5**: The video presents a sequence of images showing a street view with a pedestrian crossing. The images feature a white truck parked on the left side of the frame, a pedestrian crossing the street, and a pedestrian crossing sign visible on the right side. The sky is clear with few clouds, and the buildings in the background have a modern architectural style. The street is lined with trees and there are multiple cars parked on the right side of the street. The pedestrian is wearing dark clothing and is seen walking from one side of the street to the other.

**GPT-4**: It appears that you've provided a series of still images which seem to be sequential frames from a video that captures a scene at a traffic intersection. The visual elements include the following: a road with multiple lanes, white lane markings, and a variety of vehicles, including cars and a truck, navigating the intersection. The vehicles are captured in various positions in the lanes, implying movement through the area. There appears to be a tree-lined median or central island, and trees and buildings can be seen in the background, contributing to an urban landscape setting. The motion behavior depicted across these frames suggests the typical flow of traffic through an intersection. Vehicles are moving in a coordinated manner, adhering to lane directions and traffic signals that are implied rather than visible in the images. The sequence likely shows vehicles accelerating and decelerating as they approach and leave the intersection in response to traffic signals or the flow of other vehicles. The spatial positioning of the vehicles changes progressively, implying a real-time capture of their movements. If the sequence was put together as a continuous video, you would see vehicles entering the frame from one side and exiting on the other, with varying speeds and some vehicles possibly turning, illustrating the dynamic and fluid nature of traffic movement within a regulated urban environment.

**Gemini-Pro-1.5**: The video is a wide-angle shot of a street intersection. The image is split into three sections, providing a panoramic view. Here's a breakdown: Visual Elements: Perspective: The three-part image creates a wide-angle perspective, simulating the view a driver might have at the intersection. The left section shows the road the (assumed) viewer is on, the center focuses on the intersection itself, and the right section provides a view of the crossing traffic. Color Palette: The video appears to be in color, but the colors are somewhat muted, giving it a realistic but slightly desaturated feel. Lighting: The lighting suggests a sunny day. The left section has a bright spot from the sun, possibly late afternoon, while the right section is more evenly lit. Static Elements: Elements like trees, buildings, lampposts, and road markings provide context and a sense of place. Dynamic Elements: Vehicles, including cars and a truck, move through the intersection, creating a dynamic scene. Narrative Elements/Motion Behavior: Traffic Flow: The video captures the typical flow of traffic at an intersection. Vehicles approach, some wait their turn while others proceed through the intersection when it's clear. Right-Hand Drive: The cars driving on the right side of the road suggest the location might be a country where right-hand drive is the standard. Implied Viewer: While there is no visible camera or person, the perspective of the shot implies the viewer is positioned within a vehicle approaching the intersection. No Sound: The absence of sound enhances the focus on the visual information and allows the viewer to interpret the scene without auditory cues. Overall Impression: The video provides a snapshot of everyday life at a busy intersection. The wide-angle perspective and focus on movement create a sense of dynamism and highlight the complexities of navigating a traffic intersection.

**Wolf**: The video presents a series of scenes depicting city streets bustling with various vehicles such as cars, trucks, motorcycles, and a person riding a bicycle. The scenes are often divided into panels, each showcasing a different vehicle in motion, including a yellow taxi, a blue car, a red car, a white truck, and a silver car. The streets are marked with white lines and are surrounded by green trees and buildings, contributing to an urban landscape setting. The video captures the typical flow of traffic through a four-way intersection, with vehicles **moving in different directions**. The vehicles are shown in various positions in the lanes, implying movement through the area. The sequence likely shows vehicles **accelerating** and **decelerating** as they approach and leave the intersection in response to traffic signals or the flow of other vehicles. The video is a wide-angle shot of the street intersection, split into three sections, providing a panoramic view. The left section shows the road the viewer is on, the center focuses on the intersection itself, and the right section provides a view of the crossing traffic. The lighting suggests a sunny day, with a bright spot from the sun on the left section and more even lighting on the right. The video captures the typical flow of traffic at an intersection, with vehicles approaching, some **waiting their turn while others proceed through the intersection when it's clear**. The cars driving on the right side of the road suggest the location might be a country where right-hand drive is the standard. The absence of sound enhances the focus on the visual information and allows the viewer to interpret the scene without auditory cues. Overall, the video provides a snapshot of everyday life at a busy intersection, highlighting the complexities of navigating a traffic intersection. The wide-angle perspective and focus on movement create a sense of dynamism.

Figure 1: Comparison of CogAgent, VILA-1.5, GPT-4, Gemini-Pro-1.5, and Wolf on Interactive Nuscenes Driving Videos, Corresponding to Figure 4 of the Paper.

Lu, CK Luk, Bert Maher, Yunjie Pan, Christian Puhrsch, Matthias Reso, Mark Saroufim, Marcos Yukio Siraichi, Helen Suk, Michael Suo, Phil Tillet, Eikan Wang, Xiaodong Wang, William Wen, Shunting Zhang, Xu Zhao, Keren Zhou, Richard Zou, Ajit Mathews, Gregory Chanan, Peng Wu, and Soumith Chintala. PyTorch 2: Faster Machine Learning Through Dynamic Python Bytecode Transformation and Graph Compilation. In *29th ACM International Conference on Architectural Support for Programming Languages and Operating Systems, Volume 2 (ASPLOS '24)*. ACM, April 2024. doi: 10.1145/3620665.3640366. URL https://pytorch.org/assets/pytorch2-2.pdf.

Tim Dettmers, Artidoro Pagnoni, Ari Holtzman, and Luke Zettlemoyer. Qlora: Efficient finetuning of quantized llms. *Advances in Neural Information Processing Systems*, 36, 2024.

Wenyi Hong, Weihan Wang, Qingsong Lv, Jiazheng Xu, Wenmeng Yu, Junhui Ji, Yan Wang, Zihan Wang, Yuxiao Dong, Ming Ding, and Jie Tang. Cogagent: A visual language model for gui agents, 2024.

Ji Lin, Jiaming Tang, Haotian Tang, Shang Yang, Xingyu Dang, and Song Han. Awq: Activation-aware weight quantization for llm compression and acceleration. *arXiv preprint arXiv:2306.00978*, 2023a.

Ji Lin, Hongxu Yin, Wei Ping, Yao Lu, Pavlo Molchanov, Andrew Tao, Huizi Mao, Jan Kautz, Mohammad Shoeybi, and Song Han. Vila: On pre-training for visual language models, 2023b.

Shuming Ma, Hongyu Wang, Lingxiao Ma, Lei Wang, Wenhui Wang, Shaohan Huang, Li Dong, Ruiping Wang, Jilong Xue, and Furu Wei. The era of 1-bit llms: All large language models are in 1.58 bits. *arXiv preprint arXiv:2402.17764*, 2024.

Abhishek Padalkar, Acorn Pooley, Ajinkya Jain, Alex Bewley, Alex Herzog, Alex Irpan, Alexander Khazatsky, Anant Rai, Anikait Singh, Anthony Brohan, et al. Open x-embodiment: Robotic learning datasets and rt-x models. *arXiv preprint arXiv:2310.08864*, 2023.

Thomas Wolf, Lysandre Debut, Victor Sanh, Julien Chaumond, Clement Delangue, Anthony Moi, Pierric Cistac, Tim Rault, Rémi Louf, Morgan Funtowicz, et al. Huggingface's transformers: State-of-the-art natural language processing. *arXiv preprint arXiv:1910.03771*, 2019.