# OpenReview forum: "Wolf: Accurate Video Captioning with a World Summarization Framework"
_ICLR.cc/2025/Conference — ICLR 2025 Conference Withdrawn Submission_

### Official Review · Reviewer_PqVf · 2024-10-29

**Soundness:** 2
**Presentation:** 2
**Contribution:** 1
**Rating:** 3
**Confidence:** 4

**Summary:**

This paper presents a framework for image and video captioning by leveraging Vision Language Models and Large Language Models. Additionally, the authors propose a new metric for evaluating caption quality and develop four datasets to facilitate analysis of their framework.

**Strengths:**

This paper constructs four datasets and performs some comparative experiments to assess the effectiveness of proposed framework.

**Weaknesses:**

1. Limited novelty: The proposed framework primarily combines existing VLM and LLM models, resulting in a straightforward pipeline that lacks innovative factors. While integration seems a little valuable, the approach doesn't introduce any new concepts or methodologies. And  as far as I know, Large Language Models (LLMs) have been widely explored for video understanding tasks. What specific advantages does your framework offer compared to approaches that involve training specialized models? What evidence supports the effectiveness of relying primarily on LLMs for this complex task, especially considering the visual and temporal aspects of video analysis?
2. Dataset cohesion concerns: This paper presents four datasets, but their relationship and consistency are unclear. They appear to be sourced from different origins with varying caption annotation methods. Additionally, the 5-second video clips from the autonomous driving dataset raise questions about data sufficiency. Furthermore, the complexity of the proposed pipeline may not be suitable for autonomous driving scenarios which require real-time processing.
3. Gaps in video captioning knowledge: Video captioning is an established field with existing evaluation metrics, contrary to the authors' claim. Initially, I expected a novel metric directly calculating the consistence or similarity of video content and generated captions. However, the proposed method still relies on comparing generated captions to human annotations. The authors might benefit from reviewing evaluation methods in previous works$ ^{[1, 2]}$. Moreover, using GPT-3.5 to rewrite captions while evaluating quality with an LLM-based metric seems potentially biased.
4. Insufficient methodological details: The paper mentions bounding boxes as input, but only the NuScenes dataset appears to provide such annotations (in 3D). The method for generating these annotations for other datasets is unclear. Additionally, analysis of the framework's complexity, particularly the recursive nature of image caption generation, is absent from the experimental section.
5. Weak experiment settings: To better illustrate the effectiveness of proposed framework, evaluation on widely-used public datasets like ActivityNet$^{[3]}$ captions and YouCookII$^{[4]}$ would be more convincing than self-collected datasets. Furthermore, the decision to fine-tune VILA-1.5 on driving scene captions to demonstrate the framework's effectiveness is questionable. Improved performance on similar, even more complex datasets seems like an expected outcome, rather than a novel insight requiring experimental validation.

[1] Streaming Dense Video Captioning. CVPR 2024.

[2] Do You Remember? Dense Video Captioning with Cross-Modal Memory Retrieval. CVPR 2024.

[3] Dense-captioning events in videos. ICCV 2017.

[4] Towards automatic learning of procedures from web instructional videos. AAAI 2018.

**Questions:**

As in weaknesses.

---

### Official Review · Reviewer_Yjtd · 2024-10-30

**Soundness:** 3
**Presentation:** 3
**Contribution:** 3
**Rating:** 8
**Confidence:** 4

**Summary:**

This paper introduces Wolf (WOrLd summarization Framework), a novel framework for accurate video captioning. The key innovation is its mixture-of-experts approach that leverages both image and video-based Vision Language Models (VLMs) to generate comprehensive and accurate video descriptions. The authors also introduce CapScore, an LLM-based metric to evaluate caption quality, and create four human-annotated benchmark datasets across autonomous driving, general scenes, and robotics domains. Experimental results show that Wolf significantly outperforms existing solutions, including both research models (VILA1.5, CogAgent) and commercial solutions (Gemini-Pro-1.5, GPT-4V). The framework can be applied to enhance video understanding, auto-labeling, and captioning tasks. The authors also establish a benchmark and leaderboard for video captioning to accelerate advancement in this field.

**Strengths:**

The paper demonstrates strong originality by introducing a novel mixture-of-experts approach that combines both image and video-based VLMs for video captioning. The quality of research is evident in comprehensive experiments across multiple domains and thorough ablation studies. The presentation is clear with well-structured methodology and detailed visualizations. The work's significance is substantial: it not only advances video captioning performance but also establishes a new evaluation metric (CapScore) and benchmark datasets. The practical applications in autonomous driving and robotics demonstrate broad impact potential. By open-sourcing their code and datasets, they contribute valuable resources to the research community.

**Weaknesses:**

1.The CapScore metric's reliability needs more validation. Although human evaluation correlation is shown on the robotics dataset (100 videos), validation on larger and more diverse datasets would strengthen its credibility.

2.The ablation study is limited to model selection without exploring other important aspects like the impact of different frame sampling rates or the effectiveness of the chain-of-thought summarization process.

**Questions:**

1.Could you provide test results of CapScore's reliability across different video lengths and complexity levels?

2.Have you considered comparing CapScore against established metrics like BLEU or METEOR?

---

### Official Review · Reviewer_7Ds6 · 2024-11-04

**Soundness:** 2
**Presentation:** 2
**Contribution:** 3
**Rating:** 3
**Confidence:** 3

**Summary:**

This paper introduces Wolf, a video summarization framework for automatic video captioning that leverages an ensemble of vision-language models (VLMs) and GPT-assisted caption summarization to generate video captions. The ensemble combines image-level and video-level captions produced by each model. For image-level captions, the framework generates a caption for each key frame by inputting the current frame along with the previous frame's caption into the VLMs. Additionally, motion captions are included to track the trajectory of moving objects based on bounding box locations. Wolf Benchmark is also introduced to evaluate the model's scene comprehension and behavior understanding by generating gt captions based on gt information, rule-based methods, human inputs, and GPT-generated text. Finally, this paper proposes CapScore, a GPT-based scoring mechanism to assess the quality and similarity of predicted captions relative to gt captions. Experimental results show that the proposed Wolf model performs effectively in video captioning on the CapScore metric, compared to baselines.

**Strengths:**

- Extensive efforts for developing an automatic video captioning pipeline and curating detailed video description benchmarks.

- The curated benchmark encompasses various domains, including autonomous driving, daily life videos, and robotic manipulation, which could support advancements in scene understanding and agent capabilities.

- In addition to publishing the paper, efforts to maintain a leaderboard and continuously improve the proposed benchmarks deserve recognition.

**Weaknesses:**

- The Wolf Benchmark does not include any verification or quality control steps for the annotated ground truth captions, raising fundamental concerns about the credibility of the ground truth used for subsequent evaluations. Additionally, in the GPT rewriting step in section 4.1.1, it is unclear why GPT-3.5 was used instead of GPT-4, which is employed in the Wolf captioning pipeline.

- Related to the first weakness, CapScore seems to be a constrained metric as it does not evaluate video-to-description alignment, instead, it performs a text-to-text comparison of predictions against potentially unverified ground truth captions via LLMs. First, the approach assumes that the ground truth captions are highly accurate (e.g., oracle), which is uncertain in the current setup. Second, LLMs function as black boxes, making their scoring less transparent and challenging to interpret. Third, the approach of evaluating captions using LLMs has already been explored [Chan et al., 2023] and should be acknowledged.

- Related to the second weakness, have the authors considered using reference-less metrics that measure 'alignment' between video and descriptions, similar to methods in the image domain such as CLIPScore [Hessel et al., 2021], VisualGPTScore, and VQAScore [Lin et al., 2024a,b]?

- The comparison across baseline methods may be inherently unfair. The Wolf captioning pipeline ensembles both image-based models (CogAgent and GPT-4V) and video-based models (VILA-1.5 and Gemini-Pro-1.5) for caption generation, making direct comparisons with individual models potentially unfair and unsurprisingly favoring Wolf's superior performance.

- The limited spacing between paragraphs and between figure/table captions and the main text significantly hinders readability. For instance, the cramped spacing between lines 184 and 185 is particularly problematic. Additional clarity issues include inconsistent referencing styles within sentence components: for example, Tian et al. (2024) in line 193 and (Chen et al., 2023b) in line 210. These references should be made consistent, preferably without parentheses as the first one. In line 228, 'ect.' should be corrected to 'etc'. Consistency in terminology is needed: either 'ground-truth' or 'ground truth' throughout the paper.

---

Assesment: Concerns regarding the quality control of ground truth captions in the Wolf benchmark, issues with the caption evaluation metric, and potentially unfair comparisons with baseline models are the primary reasons for the score recommendation. Additionally, the paper's clarity, particularly its readability, requires further improvement.

---

References

[Chan et al., 2023] CLAIR: Evaluating Image Captions with Large Language Models, in EMNLP 2023.

[Hessel et al., 2021] CLIPScore: A Reference-free Evaluation Metric for Image Captioning, in EMNLP 2021.

[Lin et al., 2024a] VisualGPTScore: Visual Generative Pre-Training Score Code Implementation, in ICML 2024.

[Lin et al., 2024b] VQAScore: Evaluating Text-to-Visual Generation with Image-to-Text Generation, in ECCV 2024.

**Questions:**

- The process for manually annotating the robot manipulation dataset in section 4.1.2 and the daily life videos in section 4.1.3 is unclear, with no details provided. Additionally, the content in these sections is disproportionately brief compared to the counterpart in section 4.1.1.

---

### Official Review · Reviewer_d7iC · 2024-11-04

**Soundness:** 2
**Presentation:** 3
**Contribution:** 2
**Rating:** 5
**Confidence:** 4

**Summary:**

Wolf is a novel video captioning framework that combines multiple vision-language models (both image-based and video-based) through a mixture-of-experts approach, using chain-of-thought prompting and summarization to generate more accurate captions.

The paper introduces CapScore (a GPT-4-based evaluation metric) and creates new benchmark datasets across driving, robotics, and general scenes.

**Strengths:**

- Wolf proposes a unique mixture-of-experts approach to video captioning by combining image and video models.
- The framework leverages the complementary strengths of different models.
- Introduction of CapScore as a new evaluation metric.

**Weaknesses:**

- Method issue: There is no justification provided why CogAgent (Hongetal.,2024), GPT-4V (Mao et al.,2023a) - image and VILA-1.5 (Lin et al.,2023c), Gemini-Pro1.5 (Team et al.,2023) - video is used together and how these complement each other.

- Evaluation metric: Limited discussion of how CapScore correlates with task performance. No in-depth comparison with existing semantic-based evaluation metrics e.g. EMScore [1] or with traditional metrics e.g. cider, meteor, etc. There is heavy reliance on GPT-4 for evaluation without addressing potential biases.

- Dataset size issue: 100 robot manipulation videos seem insufficient for meaningful conclusions for robotics.

- While the paper acknowledges computational efficiency concerns, no concrete benchmarks or comparisons of computational costs are provided.

[1] Shi, Yaya, et al. "EMScore: Evaluating video captioning via coarse-grained and fine-grained embedding matching." Proceedings of the IEEE/CVF conference on computer vision and pattern recognition. 2022.

**Questions:**

- How do you handle cases where models (CogAgent (Hongetal.,2024), GPT-4V (Mao et al.,2023a) - image and VILA-1.5 (Lin et al.,2023c), Gemini-Pro1.5 (Team et al.,2023) - video) produce conflicting captions?
- How do you ensure temporal consistency when combining captions from sequential frames?
- What is the optimal sampling rate for keyframes? Why two frames per second?
- Why not include traditional or semantics-based metrics alongside CapScore?
- How do you handle real-time video captioning requirements?
- If you only do object detection on the sequence of frames and provide that to an LLM, will it model the temporal sequence?

---

### Note · Authors · 2024-11-14

I have read and agree with the venue's withdrawal policy on behalf of myself and my co-authors.